# SELF: LEARNING TO FILTER NOISY LABELS WITH SELF-ENSEMBLING

**Duc Tam Nguyen** [*], **Chaithanya Kumar Mummadi** [*†], **Thi Phuong Nhung Ngo** [†],
**Thi Hoai Phuong Nguyen** [‡], **Laura Beggel** [†], **Thomas Brox** [†]

## ABSTRACT

Deep neural networks (DNNs) have been shown to over-fit a dataset when being trained with noisy labels for a long enough time. To overcome this problem, we present a simple and effective method *self-ensemble label filtering (SELF)* to progressively filter out the wrong labels during training. Our method improves the task performance by gradually allowing supervision only from the potentially non-noisy (clean) labels and stops learning on the filtered noisy labels. For the filtering, we form running averages of predictions over the entire training dataset using the network output at different training epochs. We show that these ensemble estimates yield more accurate identification of inconsistent predictions throughout training than the single estimates of the network at the most recent training epoch. While filtered samples are removed entirely from the supervised training loss, we dynamically leverage them via semi-supervised learning in the unsupervised loss. We demonstrate the positive effect of such an approach on various image classification tasks under both symmetric and asymmetric label noise and at different noise ratios. It substantially outperforms all previous works on noise-aware learning across different datasets and can be applied to a broad set of network architectures.

## 1 INTRODUCTION

The acquisition of large quantities of a high-quality human annotation is a frequent bottleneck in applying DNNs. There are two cheap but imperfect alternatives to collect annotation at large scale: crowdsourcing from non-experts and web annotations, particularly for image data where the tags and online query keywords are treated as valid labels. Both these alternatives typically introduce noisy (wrong) labels. While Rolnick et al. (2017) empirically demonstrated that DNNs can be surprisingly robust to label noise under certain conditions, Zhang et al. (2017) has shown that DNNs have the capacity to memorize the data and will do so eventually when being confronted with too many noisy labels. Consequently, training DNNs with traditional learning procedures on noisy data strongly deteriorates their ability to generalize – a severe problem. Hence, limiting the influence of label noise is of great practical importance.

A common approach to mitigate the negative influence of noisy labels is to eliminate them from the training data and train deep learning models just with the clean labels (Frénay & Verleysen, 2013). Employing semi-supervised learning can even counteract the noisy labels (Laine & Aila, 2016; Luo et al., 2018). However, the decision which labels are noisy and which are not is decisive for learning robust models. Otherwise, unfiltered noisy labels still influence the (supervised) loss and affect the task performance as in these previous works. They use the entire label set to compute the loss and severely lack a mechanism to identify and filter out the erroneous labels from the labels set.

In this paper, we propose a *self-ensemble label filtering (SELF)* framework that identifies potentially noisy labels during training and keeps the network from receiving supervision from the filtered noisy labels. This allows DNNs to gradually focus on learning from undoubtedly correct samples even with an extreme level of noise in the labels (e.g., 80% noise ratio) and leads to improved performance as the supervision become less noisy. The key contribution of our work is progressive filtering, i.e.,

---

[*]Computer Vision Group, University of Freiburg, Germany
[†]Bosch Center for AI, Bosch GmbH, Germany
[‡]Karlsruhe Institute of Technology, Germany

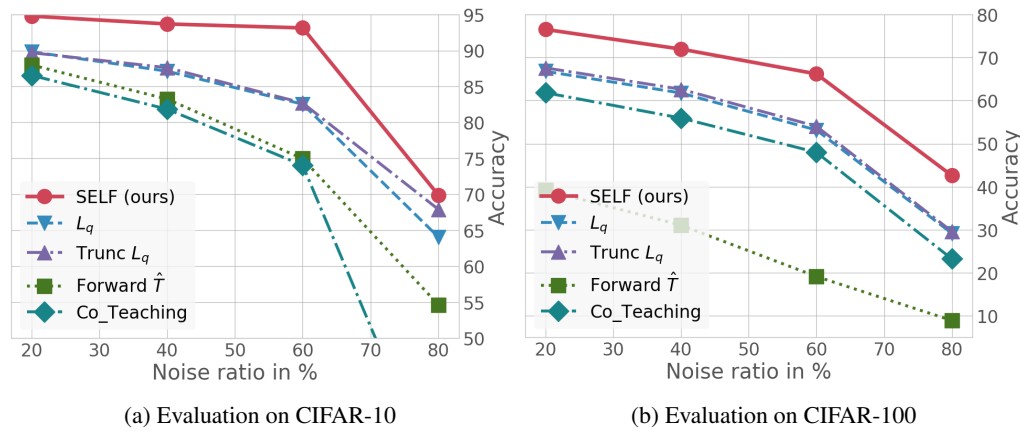

(a) Evaluation on CIFAR-10        (b) Evaluation on CIFAR-100

Figure 1: Comparing the performance of *SELF* with previous works for learning under different (symmetric) label noise ratios on the (a) CIFAR-10 & (b) CIFAR-100 datasets. *SELF* retains higher robust classification accuracy at all noise levels.

leverage the knowledge provided in the network's output over different training iterations to form a consensus of predictions *(self-ensemble predictions)* to progressively identify and filter out the noisy labels from the labeled data.

When learning under label noise, the network receives noisy updates and hence fluctuates strongly. Such conduct of training would impede to learn stable neural representations and further mislead the consensus of the predictions. Therefore, it is essential to incorporate a model with stable training behavior to obtain better estimates from the consensus. Concretely, we employ the semi-supervised technique as a backbone to our framework to stabilize the learning process of the model. Correctly, we maintain the running average model, such as proposed by Tarvainen & Valpola (2017), a.k.a. the Mean-Teacher model. This model ensemble learning provides a more stable supervisory signal than the noisy model snapshots and provides a stable ground for progressive filtering to filter out potential noisy labels. Note that this is different from just a mere combination of semi-supervised techniques with a noisy label filtering method.

We call our approach *self-ensemble label filtering (SELF)* - that establishes model ensemble learning as a backbone to form a solid consensus of the self-ensemble predictions to filter out the noisy labels progressively. Our framework allows to compute supervised loss on cleaner subsets rather than the entire noisy labeled data as in previous works. It further leverages the entire dataset, including the filtered out erroneous samples in the unsupervised loss. To best of our knowledge, we are the first to identify and propose self-ensemble as a principled technique against learning under noisy labels.

Our motivation stems from the observation that DNNs start to learn from easy samples in initial phases and gradually adapt to hard ones during training. When trained on wrongly labeled data, DNNs learn from clean labels at ease and receive inconsistent error signals from the noisy labels before over-fitting to the dataset. The network's prediction is likely to be consistent on clean samples and inconsistent or oscillates strongly on wrongly labeled samples over different training iterations. Based on this observation, we record the outputs of a single network made on different training epochs and treat them as an ensemble of predictions obtained from different individual networks. We call these ensembles that are evolved from a single network *self-ensemble predictions*. Subsequently, we identify the correctly labeled samples via the agreement between the provided label set and our running average of *self-ensemble predictions*. The samples of ensemble predictions that agree with the provided labels are likely to be consistent and treated as clean samples.

In summary, our *SELF* framework stabilizes the training process and improves the generalization ability of DNNs. We evaluate the proposed technique on image classification tasks using CI-FAR10, CIFAR100 & ImageNet. We demonstrate that SELF consistently outperforms the existing approaches on asymmetric and symmetric noise at all noise levels, as shown in Fig. 1. Besides, *SELF* remains robust towards the choice of the network architecture. Our work is transferable to other tasks without the need to modify the architecture or the primary learning objective.

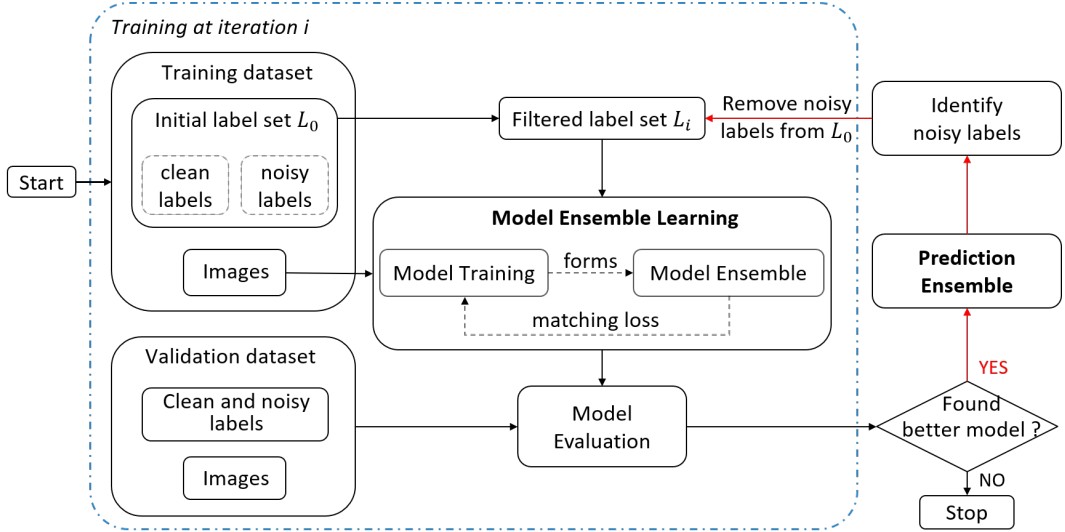

Figure 2: Overview of the *self-ensemble label filtering (SELF)* framework. The model starts in iteration 0 with training from the noisy label set. During training, the model maintains a self-ensemble, a running average of itself (Tarvainen & Valpola, 2017) to provide a stable learning signal. Also, the model collects a self-ensemble prediction (moving-average) for the subsequent filtering. Once the best model is found, these predictions identify and filter out noisy labels using the original label set $L_0$. The model performs this progressive filtering until there is no more better model. For details see Algorithm 1.

## 2 SELF-ENSEMBLE LABEL FILTERING

### 2.1 OVERVIEW

Fig. 2 shows an overview of our proposed approach. In the beginning, we assume that the labels of the training set are noisy. The model attempts to identify correct labels progressively using self-forming ensembles of models and predictions. Since wrong labels cause strong fluctuations in the model's predictions, using ensembles is a natural way to counteract noisy labels.

Concretely, in each iteration, the model learns from a detected set of potentially correct labels and maintains a running average of model snapshots (realized by the Mean Teacher model Tarvainen & Valpola (2017)). This ensemble model is evaluated on the entire dataset and provides an additional learning signal for training the single models. Additionally, our framework maintains the running-average of the model's predictions for the filtering process. The model is trained until we find the best model w.r.t. the performance on the validation set (e.g., by early-stopping). The set of correct labels is detected based on the strategy defined in Sec. 2.2. In the next iteration, we again use all data and the new filtered label set as input for the model training. The iterative training procedure stops when no better model can be found. In the following, we give more details about the combination of this training and filtering procedure.

### 2.2 PROGRESSIVE LABEL FILTERING

**Progressive detection of correctly labeled samples** Our framework Self-Ensemble Label Filtering (Algorithm 1) focuses on the detection of certainly correct labels from the provided label set $L_0$. In each iteration $i$, the model is trained using the label set of potentially correct labels $L_i$. At the end of each iteration, the model determines the next correct label set $L_{i+1}$ using the filtering strategy described in 2.2 The model stops learning when no improvement was achieved after training on the refined label set $L_{i+1}$.

In other words, in each iteration, the model attempts to learn from the easy, in some sense, *obviously* correct labels. However, learning from easy samples also affects similar but harder samples from the same classes. Therefore, by learning from these easy samples, the network can gradually distinguish between hard and wrongly-labeled samples.

---

**Algorithm 1** *SELF*: Self-Ensemble Label Filtering pseudocode

---

**Require:** $\mathcal{D}_{train}$ = noisy labeled training set
**Require:** $\mathcal{D}_{val}$ = noisy labeled validation set
**Require:** $(x, y)$ = training stimuli and label
**Require:** $\alpha$ = ensembling momentum, $0 \leq \alpha \leq 1$

$\quad i \leftarrow 0$            ▷ counter to track iterations
$\quad M_i \leftarrow train(D_{train}, D_{val})$      ▷ initial Mean-Teacher ensemble model training
$\quad M_{best} \leftarrow M_i$          ▷ set initial model as best model
$\quad \overline{z}_i \leftarrow 0$           ▷ initialize ensemble predictions of all samples
                (ignored sample index for simplicity)

$\quad$ **while** $acc(M_i, \mathcal{D}_{val}) \geq acc(M_{best}, D_{val})$ **do**   ▷ iterate until no best model is found on $\mathcal{D}_{val}$
$\quad\quad M_{best} \leftarrow M_i$        ▷ save the best model
$\quad\quad \mathcal{D}_{filter} \leftarrow \mathcal{D}_{train}$       ▷ set filtered dataset as initial label set
$\quad\quad i \leftarrow i + 1$
$\quad\quad$ **for** $(x, y)$ in $\mathcal{D}_{filter}$ **do**
$\quad\quad\quad \hat{z}_i \leftarrow M_{best}(x)$       ▷ evaluate model output $\hat{z}_i$
$\quad\quad\quad \overline{z}_i \leftarrow \alpha\overline{z}_{i-1} + (1 - \alpha)\hat{z}_i$    ▷ accumulate ensemble predictions $\overline{z}_i$
$\quad\quad\quad$ **if** $y \neq argmax(\overline{z}_i)$ **then**    ▷ verify agreement of ensemble predictions & label
$\quad\quad\quad\quad y \leftarrow \emptyset$ in $\mathcal{D}_{filter}$     ▷ identify it as noisy label & remove from label set
$\quad\quad\quad$ **end if**
$\quad\quad$ **end for**
$\quad\quad M_i \leftarrow train(D_{filter}, D_{val})$    ▷ train Mean-Teacher model on filtered label set
$\quad$ **end while**
$\quad$ **return** $M_{best}$

---

Our framework does not focus on repairing all noisy labels. Although the detection of wrong labels is sometimes easy, finding their correct hidden label might be extremely challenging in case of having many classes. If the noise is sufficiently random, the set of correct labels will be representative to achieve high model performance. Further, in our framework, the label filtering is performed on the *original* label set $L_0$ from iteration 0. Clean labels erroneously removed in an earlier iteration (e.g., labels of hard to classify samples) can be reconsidered for model training again in later iterations.

**Filtering strategy**   The model can determine the set of potentially correct labels $L_i$ based on agreement between the label $y$ and its maximal likelihood prediction $\hat{y}|x$ with $L_i = \{(y, x) \mid \hat{y}_x = y; \forall(y, x) \in L_0\}$. $L_0$ is the label set provided in the beginning, $(y, x)$ are the samples and their respective noisy labels in the iteration $i$. In other words, the labels are only used for supervised training if in the current epoch, the model predicts the respective label to be the correct class with the highest likelihood. In practice, our framework does not use $\hat{y}(x)$ of model snapshots for filtering but a moving-average of the ensemble models and predictions to improve the filtering decision.

## 2.3 SELF-ENSEMBLE LEARNING

The model's predictions for noisy samples tend to fluctuate. For example, take a cat wrongly labeled as a tiger. Other cat samples would encourage the model to predict the given cat image as a cat. Contrary, the wrong label *tiger* regularly pulls the model back to predict the *cat* as a tiger. Hence, using the model's predictions gathered in one single training epoch for filtering is sub-optimal. Therefore, in our framework *SELF*, our model relies on ensembles of models and predictions.

**Model ensemble with Mean Teacher**   A natural way to form a model ensemble is by using an exponential running average of model snapshots (Fig. 3a). This idea was proposed in Tarvainen & Valpola (2017) for semi-supervised learning and is known as the Mean Teacher model. In our framework, both the mean teacher model and the normal model are evaluated on *all* data to preserve the consistency between both models. The consistency loss between student and teacher output distribution can be realized with Mean-Square-Error loss or Kullback-Leibler-divergence. More details for training with the model ensemble can be found in Appendix A.1

**Prediction ensemble**   Additionally, we propose to collect the sample predictions over multiple training epochs: $\overline{z}_j = \alpha\overline{z}_{j-1} + (1 - \alpha)\hat{z}_j$ , whereby $\overline{z}_j$ depicts the moving-average prediction of sample $k$ at epoch $j$, $\alpha$ is a momentum, $\hat{z}_j$ is the model prediction for sample $k$ in epoch $j$. This scheme is displayed in Fig. 3b. For each sample, we store the moving-average predictions, accumulated over the past iterations. Besides having a more stable basis for the filtering step, our proposed procedure also leads to negligible memory and computation overhead.

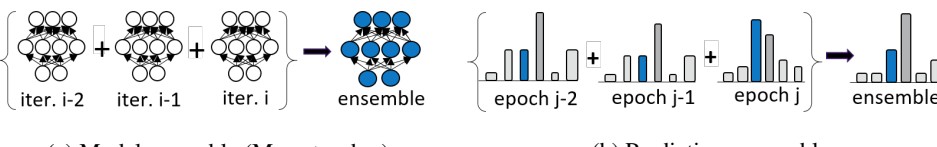

(a) Model ensemble (Mean teacher)       (b) Predictions ensemble

Figure 3: Maintaining the (a) model and (b) predictions ensembles is very effective against noisy model updates. These ensembles are self-forming during the training process as a moving-average of (a) model snapshots or (b) class predictions from previous training steps.

Further, due to continuous training of the best model from the previous model, computation time can be significantly reduced, compared to re-training the model from scratch. On the new filtered dataset, the model must only slowly adapt to the new noise ratio contained in the training set. Depending on the computation budget, a maximal number of iterations for filtering can be set to save time.

## 3 RELATED WORKS

Reed et al. (2014); Azadi et al. (2015) performed early works on learning robustly under label noise for deep neural networks. Recently, Rolnick et al. (2017) have shown for classification that deep neural networks come with natural robustness to label noise following a particular random distribution. No modification of the network or the training procedure is required to achieve this robustness. Following this insight, our framework *SELF* relies on this *natural robustness* to kickstart the self-ensemble filtering process to extend the robust behavior to more challenging scenarios.

Laine & Aila (2016); Luo et al. (2018) proposed to apply semi-supervised techniques on the data to counteract noise. These and other semi-supervised learning techniques learn from a static, initial set of noisy labels and have no mechanisms to repair labels. Therefore, the supervised losses in their learning objective are typically high until the model strongly overfits to the label noise. Compared to these works, our framework performs a variant of self-supervised label corrections. The network learns from a dynamic, variable set of labels, which is determined by the network itself. Progressive filtering allows the network to (1) focus on a label set with a significantly lower noise ratio and (2) repair wrong decisions made by itself in an earlier iteration.

Other works assign weights to potentially wrong labels to reduce the learning signal (Jiang et al., 2017; Ren et al., 2018; Jenni & Favaro, 2018). These approaches tend to assign less extreme weights or hyperparameters that are hard to set. Since the typical classification loss is highly non-linear, a lower weight might still lead to learning from wrong labels. Compared to these works, the samples in *SELF* only receive extreme weights: either they are zero or one. Further, *SELF* focuses only on self-detecting the correct samples, instead of repairing the wrong labels. Typically, the set of correct samples are much easier to detect and are sufficiently representative to achieve high performance.

Han et al. (2018b); Jiang et al. (2017) employ two collaborating and simultaneously learning networks to determine which samples to learn from and which not. However, the second network is free in its predictions and hence hard to tune. Compared to these works, we use ensemble learning as a principled approach to counteract model fluctuations. In *SELF*, the second network is extremely restricted and is only composed of running averages of the first network. To realize the second network, we use the mean-teacher model (Tarvainen & Valpola, 2017) as a backbone. Compared to their work, our self-ensemble label filtering gradually detects the correct labels and learns from them, so the label set is variable. Further, we do use not only model ensembles but also an ensemble of predictions to detect correct labels.

Other works modify the primary loss function of the classification tasks. Patrini et al. (2017) estimates the noise transition matrix to correct the loss, Han et al. (2018a) uses human-in-the-loop, Zhang & Sabuncu (2018); Thulasidasan et al. (2019) propose other forms of cross-entropy losses. The loss modification impedes the transfer of these ideas to other tasks than classification. Compared to these works, our framework *SELF* does not modify the primary loss. However, many tasks rely on the presence of clean labels such as anomaly detection (Nguyen et al., 2019a) or self-supervised and unsupervised learning (Nguyen et al., 2019b). The progressive filtering procedure and self-ensemble learning proposed are also applicable in these tasks to counteract noise effectively.

Table 1: Comparison of classification accuracy when learning under uniform label noise on CIFAR-10 and CIFAR-100. Following previous works, we compare two evaluation scenarios: with a noisy validation set (top) and with 1000 clean validation samples (bottom). The best model is marked in bold. Having a small clean validation set improves the model but is not necessary.

| | CIFAR-10 | | | CIFAR-100 | | |
|---|---|---|---|---|---|---|
| NOISE RATIO | 40% | 60% | 80% | 40% | 60% | 80 % |
| USING NOISY VALIDATION SET | | | | | | |
| REED-HARD (REED ET AL., 2014) | 69.66 | - | - | 51.34 | - | - |
| S-MODEL (GOLDBERGER & BEN-REUVEN, 2016) | 70.64 | - | - | 49.10 | - | - |
| OPEN-SET WANG ET AL. (2018) | 78.15 | - | - | - | - | - |
| RAND. WEIGHTS (REN ET AL., 2018) | 86.06 | - | - | 58.01 | - | - |
| BI-LEVEL-MODEL (JENNI & FAVARO, 2018) | 89.00 | - | 20.00 | 61.00 | - | 13.00 |
| MENTORNET (JIANG ET AL., 2017) | 89.00 | - | 49.00 | 68.00 | - | 35.00 |
| $L_q$ (ZHANG & SABUNCU, 2018) | 87.13 | 82.54 | 64.07 | 61.77 | 53.16 | 29.16 |
| TRUNC $L_q$(ZHANG & SABUNCU, 2018) | 87.62 | 82.70 | 67.92 | 62.64 | 54.04 | 29.60 |
| FORWARD $\hat{T}$ (PATRINI ET AL., 2017) | 83.25 | 74.96 | 54.64 | 31.05 | 19.12 | 08.90 |
| CO-TEACHING (HAN ET AL., 2018B) | 81.85 | 74.04 | 29.22 | 55.95 | 47.98 | 23.22 |
| D2L (MA ET AL., 2018) | 83.36 | 72.84 | - | 52.01 | 42.27 | |
| SL (WANG ET AL., 2019) | 85.34 | 80.07 | 53.81 | 53.69 | 41.47 | 15.00 |
| JOINTOPT (TANAKA ET AL., 2018) | 83.27 | 74.39 | 40.09 | 52.88 | 42.64 | 18.46 |
| SELF (OURS) | **93.70** | **93.15** | **69.91** | **71.98** | **66.21** | **42.09** |
| USING CLEAN VALIDATION SET (1000 IMAGES) | | | | | | |
| DAC (THULASIDASAN ET AL., 2019) | 90.93 | 87.58 | 70.80 | 68.20 | 59.44 | 34.06 |
| MENTORNET (JIANG ET AL., 2017) | 78.00 | - | - | 59.00 | - | - |
| RAND. WEIGHTS (REN ET AL., 2018) | 86.55 | - | - | 58.34 | - | - |
| REN ET AL (REN ET AL., 2018) | 86.92 | - | - | 61.31 | - | - |
| SELF* (OURS) | **95.10** | **93.77** | **79.93** | **74.76** | **68.35** | **46.43** |

## 4 EVALUATION

### 4.1 EXPERIMENTS DESCRIPTIONS

#### 4.1.1 STRUCTURE OF THE ANALYSIS

We evaluate our approach on CIFAR-10, CIFAR-100, an ImageNet-ILSVRC on different noise scenarios. For CIFAR-10, CIFAR-100, and ImageNet, we consider the typical situation with symmetric and asymmetric label noise. In the case of the symmetric noise, a label is randomly flipped to another class with probability $p$. Following previous works, we also consider label flips of semantically similar classes on CIFAR-10, and pair-wise label flips on CIFAR-100. Finally, we perform studies on the choice of the network architecture and the ablation of the components in our framework. Tab. 6 (Appendix) shows the in-deep analysis of semi-supervised learning strategies combined with recent works. Overall, the proposed framework SELF outperforms all these combinations.

#### 4.1.2 COMPARISONS TO PREVIOUS WORKS

We compare our work to previous methods from Reed-Hard (Reed et al., 2014), S-model (Goldberger & Ben-Reuven, 2016), Wang et al. (2018), Rand. weights (Ren et al., 2018), Bi-level-model (Jenni & Favaro, 2018), D2L (Ma et al., 2018), SL (Wang et al., 2019), $L_q$ (Zhang & Sabuncu, 2018), Trunc $L_q$ (Zhang & Sabuncu, 2018), Forward $\hat{T}$ (Patrini et al., 2017), DAC (Thulasidasan et al., 2019), Random reweighting (Ren et al., 2018), and Learning to reweight (Ren et al., 2018). For co-teaching (Han et al., 2018b), MentorNet (Jiang et al., 2017), JointOpt (Tanaka et al., 2018), the source codes are available and hence used for evaluation.

(Ren et al., 2018) and DAC (Thulasidasan et al., 2019) considered the setting of having a small clean validation set of 1000 and 5000 images respectively. For comparison purposes, we also experiment with a small clean set of 1000 images additionally. Further, we abandon oracle experiments or methods using additional information to keep the evaluation comparable. For instance, Forward $T$ (Patrini et al., 2017) uses the *true* underlying confusion matrix to correct the loss. This information is neither known in typical scenarios nor used by other methods.

Table 2: Asymmetric noise on CIFAR-10, CIFAR-100. All methods use Resnet34. **CIFAR-10**: flip TRUCK → AUTOMOBILE, BIRD → AIRPLANE, DEER → HORSE, CAT↔DOG with prob. $p$. **CIFAR-100**: flip class $i$ to $(i+1)\%100$ with prob. $p$. *SELF* retains high performances across all noise ratios and outperforms all previous works.

| Noise ratio | Cifar-10 | | | | CIFAR-100 | | | |
|---|---|---|---|---|---|---|---|---|
| | 10% | 20% | 30% | 40% | 10% | 20% | 30% | 40% |
| CCE | 90.69 | 88.59 | 86.14 | 80.11 | 66.54 | 59.20 | 51.40 | 42.74 |
| MAE | 82.61 | 52.93 | 50.36 | 45.52 | 13.38 | 11.50 | 08.91 | 08.20 |
| Forward $\hat{T}$ | 90.52 | 89.09 | 86.79 | 83.55 | 45.96 | 42.46 | 38.13 | 34.44 |
| $L_q$ | 90.91 | 89.33 | 85.45 | 76.74 | 68.36 | 66.59 | 61.45 | 47.22 |
| Trunc $L_q$ | 90.43 | 89.45 | 87.10 | 82.28 | 68.86 | 66.59 | 61.87 | 47.66 |
| SL | 88.24 | 85.36 | 80.64 | - | 65.58 | 65.14 | 63.10 | - |
| JointOpt | 90.12 | 89.45 | 87.18 | 87.97 | 69.61 | 68.94 | 63.99 | 53.71 |
| SELF (Ours) | **93.75** | **92.76** | **92.42** | **89.07** | **72.45** | **70.53** | **65.09** | **53.83** |

Whenever possible, we adopt the reported performance from the corresponding publications. The testing scenarios are kept as similar as possible to enable a fair comparison. All tested scenarios use a noisy validation set with the same noise distribution as the training set unless stated otherwise. All model performances are reported on the *clean test set*.

Table 3: Effect of the choice of network architecture on classification accuracy on CIFAR-10 & -100 with uniform label noise. *SELF* is compatible with all tested architectures. Here * represents baseline accuracy of the architectures that are trained on fully supervised setting at 0% label noise.

| | Cifar-10 | | CIFAR-100 | |
|---|---|---|---|---|
| Resnet101 | 93.89* | | 81.14* | |
| Noise | 40% | 80% | 40% | 80% |
| Mentornet | 89.00 | 49.00 | 68.00 | 35.00 |
| Co-T. | 62.58 | 21.79 | 39.58 | 16.79 |
| SELF | **92.77** | **64.52** | **69.00** | **39.73** |

| | Cifar-10 | | CIFAR-100 | |
|---|---|---|---|---|
| WRN 28-10 | 96.21* | | 81.02* | |
| Noise | 40% | 80% | 40% | 80% |
| Mentornet | 88.7 | 46.30 | 67.50 | 30.10 |
| Reweight | 86.02 | - | 58.01 | - |
| SELF | **93.34** | **67.41** | **72.48** | **42.06** |

| | Cifar-10 | | CIFAR-100 | |
|---|---|---|---|---|
| Resnet34 | 93.5* | | 76.76* | |
| Noise | 40% | 80% | 40% | 80% |
| $L_q$ | 87.13 | 64.07 | 61.77 | 29.16 |
| Trunc $L_q$ | 87.62 | **67.92** | 62.64 | 29.60 |
| Forward $\hat{T}$ | 83.25 | 54.64 | 31.05 | 8.90 |
| SELF | **91.13** | 63.59 | **66.71** | **35.56** |

| | Cifar-10 | | CIFAR-100 | |
|---|---|---|---|---|
| Resnet26 | 96.37* | | 81.20* | |
| Noise | 40% | 80% | 40% | 80% |
| Co-T. | 81.85 | 29.22 | 55.95 | 23.22 |
| SELF | **93.70** | **69.91** | **71.98** | **42.09** |

### 4.1.3 Networks configuration and training

For the basic training of self-ensemble model, we use the Mean Teacher model (Tarvainen & Valpola, 2017) available on GitHub [1] . The students and teacher networks are residual networks (He et al., 2016) with 26 layers with Shake-Shake-regularization (Gastaldi, 2017). We use the PyTorch (Paszke et al., 2017) implementation of the network and keep the training settings close to (Tarvainen & Valpola, 2017). The network is trained with Stochastic Gradient Descent. In each filtering iteration, the model is trained for a maximum of 300 epochs, with patience of 50 epochs. For more training details, see the appendix.

### 4.2 Experiments results

### 4.2.1 Symmetric label noise

**CIFAR-10 and 100**   Results for typical uniform noise scenarios with noise ratios on CIFAR-10 and CIFAR-100 are shown in Tab. 1. More results are visualized in Fig. 1a (CIFAR-10) and Fig. 1b (CIFAR-100). Our approach *SELF* performs robustly in the case of lower noise ratios with up to 60% and outperforms previous works. Although a strong performance loss occurs at 80% label noise,

[1]https://github.com/CuriousAI/mean-teacher

Table 4: Classification accuracy on clean ImageNet validation dataset. The models are trained at 40% label noise and the best model is picked based on the evaluation on noisy validation data. Mentornet shows the best previously reported results. Mentornet* is based on Resnet-101. We chose the smaller Resnext50 model to reduce the run-time.

| Accurracy | Resnext18 | | Resnext50 | |
| | P@1 | P@5 | P@1 | P@5 |
| --- | --- | --- | --- | --- |
| Mentornet* | - | - | 65.10 | 85.90 |
| ResNext | 50.6 | 75.99 | 56.25 | 80.90 |
| Mean-T. | 58.04 | 81.82 | 62.96 | 85.72 |
| SELF (Ours) | **66.92** | **86.65** | **71.31** | **89.92** |

Table 5: Ablation study on CIFAR-10 and CIFAR-100. The Resnet baseline was trained on the full noisy label set. Adding progressive filtering improves over this baseline. The Mean Teacher maintains an ensemble of model snapshots, which helps counteract noise. Having progressive filtering and model ensembles (-MVA-pred.) makes the model more robust but still fails at 80% noise. The full *SELF* framework additionally uses the prediction ensemble for detection of correct labels.

| | CIFAR-10 | | CIFAR-100 | |
| NOISE RATIO | 40% | 80% | 40% | 80% |
| --- | --- | --- | --- | --- |
| RESNET26 | 83.20 | 41.37 | 53.18 | 19.92 |
| FILTERING | 87.35 | 49.58 | 61.40 | 23.42 |
| MEAN-T. | 93.70 | 52.50 | 65.85 | 26.31 |
| - MVA-PRED. | 93.77 | 57.40 | 71.69 | 38,61 |
| SELF (OURS) | **93.70** | **69.91** | **71.98** | **42.09** |

*SELF* still outperforms most of the previous approaches. The experiment *SELF*\* using a 1000 clean validation images shows that the performance loss mostly originates from the progressive filtering relying too strongly on the extremely noisy validation set.

**ImageNet-ILSVRC** Tab. 4 shows the precision@1 and @5 of various models, given 40% label noise in the training set. Our networks are based on ResNext18 and Resnext50. Note that MentorNet (Jiang et al., 2017) uses Resnet101 (P@1: **78.25**) (Goyal et al., 2017), which has higher performance compared to Resnext50 (P@1: **77.8**) (Xie et al., 2017) on the standard ImageNet validation set.

Despite the weaker model, *SELF* (ResNext50) surpasses the best previously reported results by more than 5% absolute improvement. Even the significantly weaker model ResNext18 outperforms MentorNet, which is based on a more powerful ResNet101 network.

### 4.2.2 ASYMMETRIC LABEL NOISE

Tab. 2 shows more challenging noise scenarios when the noise is not class-symmetric and uniform. Concretely, labels are flipped among semantically similar classes such as CAT and DOG on CIFAR-10. On CIFAR-100, each label is flipped to the next class with a probability $p$. In these scenarios, our framework *SELF* also retains high performance and only shows a small performance drop at 40% noise. The high label noise resistance of our framework indicates that the proposed self-ensemble filtering process helps the network identify correct samples, even under extreme noise ratios.

### 4.2.3 EFFECTS OF DIFFERENT ARCHITECTURES

Previous works utilize a various set of different architectures, which hinders a fair comparison. Tab. 3 shows the performance of our framework *SELF* compared to previous approaches. *SELF* outperforms other works in all scenarios except for CIFAR-10 with 80% noise. Typical robust learning approaches lead to significant accuracy losses at 40% noise, while *SELF* still retains high performance. Further, note that *SELF* allows the network's performance to remain consistent across the different underlying architectures.

### 4.2.4 ABLATION STUDY

Tab. 5 shows the importance of each component in our framework. See Fig. 4a, Fig. 4b for experiments on more noise ratios. As expected, the Resnet-baseline rapidly breaks down with increasing noise ratios. Adding self-supervised filtering increases the performance slightly in lower noise ratios. However, the model has to rely on extremely noisy snapshots. Contrary, using a model ensemble alone such as in Mean-Teacher can counteract noise on the noisy dataset CIFAR-10. On the more challenging CIFAR-100, however, the performance decreases strongly. With self-supervised filtering and model ensembles, *SELF* (without MVA-pred) is more robust and only impairs performance at 80% noise. The last performance boost is given by using moving-average predictions so that the network can reliably detect correctly labeled samples gradually.

Fig. 4 shows the ablation experiments on more noise ratios. The analyses shows that each component in SELF is essential for the model to learn robustly.

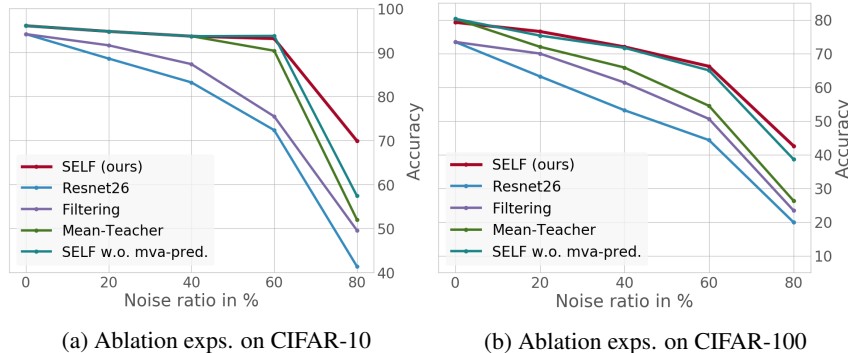

(a) Ablation exps. on CIFAR-10                 (b) Ablation exps. on CIFAR-100

Figure 4: Ablation study on the importance of the components in our framework *SELF*, evaluated on (a) Cifar-10 and (b) Cifar-100 with uniform noise. Please refer Tab. 5 for details of components.

Table 6: Analysis of semi-supervised learning (SSL) strategies: entropy learning, mean-teacher combined with recent works. Our progressive filtering strategy is shown to be effective and performs well regardless of the choice of the semi-supervised learning backbone. Overall, the proposed method SELF outperforms all these combinations. Best model in each SSL-category is marked in bold. Running mean-teacher+ co-teaching using the same configuration is not possible due to memory constraints.

| | Cifar-10 | | | Cifar-100 | | |
|---|---|---|---|---|---|---|
| Noise ratio | 40% | 60% | 80% | 40% | 60% | 80% |
| BASELINE MODELS | | | | | | |
| Resnet26 (Gastaldi, 2017) | 83.20 | 72.35 | 41.37 | 53.18 | 44.31 | 19.92 |
| Co-Teaching (Han et al., 2018b) | 81.85 | 74.04 | 29.22 | 55.95 | 47.98 | 23.22 |
| JointOpt (Tanaka et al., 2018) | 83.27 | 74.39 | 40.09 | 52.88 | 42.64 | 18.46 |
| Progressive Filtering (ours) | **87.35** | **75.47** | **49.58** | **61.40** | **50.60** | **23.42** |
| SEMI-SUPERVISED LEARNING WITH ENTROPY LEARNING | | | | | | |
| Entropy | 79.13 | 85.98 | 46.93 | 54.65 | 41.34 | 21.29 |
| Entropy + Co-Teaching | 84.94 | 74.28 | 35.16 | 55.68 | 43.52 | 20.5 |
| Entropy + Joint-Opt | 84.44 | 75.86 | 39.16 | 56.73 | 43.27 | 17.24 |
| Entropy+Filtering (ours) | **90.04** | **83.88** | **52.46** | **59.97** | **46.45** | **23.53** |
| SEMI-SUPERVISED LEARNING WITH MEAN-TEACHER | | | | | | |
| Mean Teacher | 93.70 | 90.40 | 52.5 | 65.85 | 54.48 | 26.31 |
| Mean-Teacher + JointOpt | 91.40 | 83.62 | 45.12 | 60.09 | 45.92 | 23.54 |
| Mean-Teacher + Filtering - *SELF* (ours) | **93.70** | **92.85** | **69.91** | **71.98** | **66.21** | **42.58** |

### 4.2.5 SEMI-SUPERVISED LEARNING FOR PROGRESSIVE FILTERING

Tab. 6 shows different semi-supervised learning strategies: entropy learning, mean-teacher combined with recent works. Note that Co-Teaching+Mean-Teacher cannot be implemented and run in the same configuration as other experiments, due to memory constraints.

The analysis indicates the semi-supervised losses mostly stabilize the baselines, compared to the model without semi-supervised learning. However, Co-teaching and JointOpt sometimes perform worse than the purely semi-supervised model. This result indicates that their proposed frameworks are not always compatible with semi-supervised losses.

The progressive filtering technique is seamlessly compatible with different semi-supervised losses. The filtering outperforms its counterparts when combined with Entropy Learning or Mean-teacher model. Overall, SELF outperforms all considered combinations.

## 5 CONCLUSION

We propose a simple and easy to implement a framework to train robust deep learning models under incorrect or noisy labels. We filter out the training samples that are hard to learn (possibly noisy labeled samples) by leveraging ensemble of predictions of the single network's output over different training epochs. Subsequently, we allow clean supervision from the non-hard samples and further leverage additional unsupervised loss from the entire dataset. We show that our framework results in DNN models with superior generalization performance on CIFAR-10, CIFAR-100 & ImageNet and outperforms all previous works under symmetric (uniform) and asymmetric noises. Furthermore, our models remain robust despite the increasing noise ratio and change in network architectures.

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

# A APPENDIX

## A.1 MEAN TEACHER MODEL FOR ITERATIVE FILTERING

We apply the Mean Teacher algorithm in each iteration $i$ in the $train(\mathcal{D}_{filter}, \mathcal{D}_{val})$ procedure as follows.

- Input: examples with potentially clean labels $D_{filter}$ from the filtering procedure. In the beginning ($i = 0$), here $D_{filter}$ refers to entire labeled dataset.
- Initialize a supervised neural network as the student model $M_i^s$.
- Initialize the Mean Teacher model $M_i^t$ as a copy of the student model with all weights detached.
- Let the loss function be the sum of normal classification loss of $M_i^s$ and the *consistency loss* between the outputs of $M_i^t$ and $M_i^t$
- Select an optimizer
- In each training iteration:
    - Update the weights of $M_i^s$ using the selected optimizer
    - Update the weights of $M_i^t$ as an exponential moving-average of the student weights
    - Evaluate performance of $M_i^s$ and $M_i^t$ over $\mathcal{D}_{val}$ to verify the early stopping criteria.
- Return the best $M_i^t$

## A.2 ASSUMPTIONS DICUSSIONS

Our method performs best when the following assumptions are hold.

**Natural robustness assumption of deep networks**   (Rolnick et al., 2017): The networks attempt to learn the easiest way to explain most of the data. SELF uses this assumption to kickstart the learning process.

**Correct samples dominate over wrongly labeled samples**   At 80% noise on CIFAR-10, the correctly labeled cats (20% out of all cat images) still dominates over samples wrongly labeled as cat ($8.\bar{8}$% for each class).

**Independence results in less overfitting**   SELF performs best if the noises on the validation set and training set are i.i.d. . SELF uses the validation data for early stopping. Hence, a high correlation of label noise between train and valid increases the chance of model overfitting.

**Sufficient label randomness assumption**   The subset of all correctly labeled samples capture all samples clusters. In fact, many works from the active learning literature show that less than 100 % of the labeled samples are required to achieve the highest model performance. SELF performs progressive expansion of the correct labels sets. At larger noise ratios, not all clusters are covered by the identified samples. Therefore on task containing many classes, e.g., CIFAR-100, the model performance decreases faster than on CIFAR-10.

The model performance reduces when these assumptions are strongly violated. Each assumption should have its own "critical" threshold for violation. A future in-depth analysis to challenge the assumptions is an interesting future research direction.

## A.3 TRAINING DETAILS

### A.3.1 CIFAR-10 AND CIFAR-100

**Dataset**   Tab. 7 shows the details of CIFAR-10 and 100 datasets in our evaluation pipeline. The validation set is contaminated with the same noise ratio as the training data unless stated otherwise.

**Network training**    For the training our model *SELF*, we use the standard configuration provided by Tarvainen & Valpola (2017) [2]. Concretely, we use the SGD-optimizer with Nesterov Sutskever et al. (2013) momentum, a learning rate of 0.05 with cosine learning rate annealing Loshchilov & Hutter (2016), a weight decay of 2e-4, max iteration per filtering step of 300, patience of 50 epochs, total epochs count of 600.

Table 7: Dataset description. Classification tasks on CIFAR-10 and CIFAR-100 with uniform noise. Note that the noise on the training and validation set is not correlated. Hence, maximizing the accuracy on the noisy set provides a useful (but noisy) estimate for the generalization ability on unseen test data.

|  | TYPE | CIFAR-10 | CIFAR-100 |
|---|---|---|---|
| TASK RESOLUTION | CLASSIFICATION | 10-WAY
32x32 | 100-WAY |
| DATA | TRAIN (NOISY)
VALID (NOISY)
TEST (CLEAN) | 45000
5000
10000 | 45000
5000
10000 |

For basic training of baselines models without semi-supervised learning, we had to set the learning rate to 0.01. In the case of higher learning rates, the loss typically explodes. Every other option is kept the same.

**Semi-supervised learning**    For the mean teacher training, additional hyperparameters are required. In both cases of CIFAR-10 and CIFAR-100, we again take the standard configuration with the consistency loss to mean-squared-error and a consistency weight: 100.0, logit distance cost: 0.01, consistency-ramp-up:5. The total batch-size is 512, with 124 samples being reserved for labeled samples, 388 for unlabeled data. Each epoch is defined as a complete processing of all unlabeled data. When training without semi-supervised-learning, the entire batch is used for labeled data.

**Data augmentation**    The data are normalized to zero-mean and standard-variance of one. Further, we use real-time data augmentation with random translation and reflection, subsequently random horizontal flip. The standard PyTorch-library provides these transformations.

### A.3.2    IMAGENET-ILSVRC-2015

**Network Training**    The network used for evaluation were ResNet He et al. (2016) and Resnext Xie et al. (2017) for training. All ResNext variants use a cardinality of 32 and base width of 4 (32x4d). ResNext models follow the same structure as their Resnet counterparts, except for the cardinality and base width.

All other configurations are kept as close as possible to Tarvainen & Valpola (2017). The initial learning rate to handle large batches Goyal et al. (2017) is set to 0.1; the base learning rate is 0.025 with a single cycle of cosine annealing.

**Semi-supervised learning**    Due to the large images, the batch size is set to 40 in total with 20/20 for labeled and unlabeled samples, respectively. We found the Kullback-divergence leads to no meaningful network training. Hence, we set the consistency loss to mean-squared-error, with a weight of 1000. We use consistency ramp up of 5 epochs to give the mean teacher more time in the beginning. Weight decay is set to 5e-5; patience is four epochs to stop training in the current filtering iteration.

**Filtering**    We filter noisy samples with the topk=5 strategy, instead of taking the maximum-likelihood (ML) prediction as on CIFAR-10 and CIFAR-100. That means the samples are kept for supervised training if their provided label lies within the top 5 predictions of the model. The main

---

[2]https://github.com/CuriousAI/mean-teacher

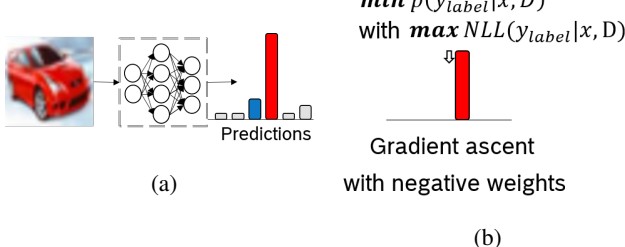

$$\boldsymbol{min}\, p(y_{label}|x,D)$$
with $\boldsymbol{max}\, NLL(y_{label}|x,\mathrm{D})$

Predictions

(a)

Gradient ascent
with negative weights

(b)

Figure 5: Simple training losses to counter label noise. (a) shows the prediction of a sample given a model. The red bar indicates the noisy label, blue the correct one. Arrows depict the magnitude of the gradients (b) Typical losses reweighting schemes are not wrong but suffer from the gradient vanishing problem. Non-linear losses such as Negative-log-likelihood are not designed for gradient ascent.

reason is that each image of ImageNet might contain multiple objects. Filtering with ML-predictions is too strict and would lead to a small recall of the detection of the correct sample.

**Data Augmentation**  For all data, we normalize the RGB-images by the mean: (0.485, 0.456, 0.406) and the standard variance (0.229, 0.224, 0.225). For training data, we perform a random rotation of up to 10 degrees, randomly resize images to 224x224, apply random horizontal flip and random color jittering. This noise is needed in regular mean-teacher training. The jittering setting are: brightness=0.4, contrast=0.4, saturation=0.4, hue=0.1. The validation data are resized to 256x256 and randomly cropped to 224x224

### A.3.3   SEMI-SUPERVISED LOSSES

For the learning of wrongly labeled samples, Fig. 6 shows the relationship between the typical reweighting scheme and our baseline push-away-loss. Typically, reweighting is applied directly to the losses with samples weights $w^{(k)}$ for each sample $k$ as shown in Eq. 4

$$\min w_i^{(k)} NLL(y_{label}^{(k)}|x^{(k)}, D) \tag{1}$$

$D$ is the dataset, $x^{(k)}$ and $y_{label}^{(k)}$ are the samples $k$ and its noisy label. $w_i^{(k)}$ is the samples weight for the sample $k$ at step $i$. Negative samples weights $w_i^{(k)}$ are often assigned to push the network away from the wrong labels. Let $w_i^{(k)} = -c_i^{(k)}$ with $c_i^{(k)} > 0$, then we have:

$$\min -c_i^{(k)} NLL(y_{label}^{(k)}|x^{(k)}, D) \tag{2}$$

Which results in:

$$\max c_i^{(k)} NLL(y_{label}^{(k)}|x^{(k)}, D) \tag{3}$$

In other words, we perform gradient ascent for wrongly labeled samples. However, the Negative-log-likelihood is not designed for gradient ascent. Hence the gradients of wrongly labeled samples vanish if the prediction is too close to the noisy label. This effect is similar to the training of Generative Adversarial Network (GAN) Goodfellow et al.. In the GAN-framework, the generator loss is not simply set to the negated version of the discriminator's loss for the same reason.

Therefore, to provide a fair comparison with our framework, we suggest the push-away-loss $L_{Push-away}(y_{label}^{(k)}, x^{(k)}, D)$ with improved gradients as follows:

$$\min \frac{1}{|Y|-1} \sum_{y, y \neq y_{label}^{(k)}} c_i^{(k)} NLL(y|x^{(k)}, D) \tag{4}$$

Whereby $Y$ is the set of all classes in the training set. This loss has improved gradients to push the model away from the potentially wrong labels.

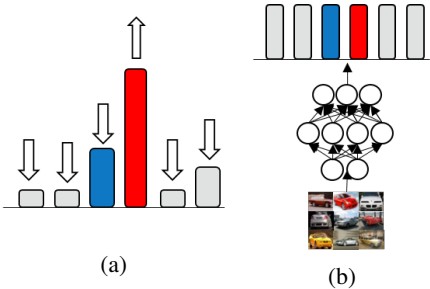

(a)

(b)

Figure 6: The entropy loss for semi-supervised learning. (a) Extreme predictions such as $[0, 1]$ are encouraged by minimizing the entropy on each prediction. (b) Additionally, maximizing the entropy of the mean prediction on the entire dataset or a large batch forces the model to balance its predictions over multiple samples.

Table 8: Accuracy of the complete removal of samples during iterative filtering on CIFAR-10 and CIFAR-100. The underlying model is the MeanTeacher based on Resnet26. When samples are completely removed from the training set, they are no longer used for either supervised-or-unsupervised learning. This common strategy from previous works leads to rapid performance breakdown.

|  | CIFAR-10 | | CIFAR-100 | |
| --- | --- | --- | --- | --- |
| NOISE RATIO | 40% | 80 % | 40% | 80 % |
| USING NOISY DATA ONLY | | | | |
| DATA REMOVAL | 93.4 | 59.98 | 68.99 | 35.53 |
| *SELF* (OURS) | **93.7** | **69.91** | **71.98** | **42.09** |
| WITH CLEAN VALIDATION SET | | | | |
| COMPL. REMOVAL | 94.39 | 70.93 | 71.86 | 36.61 |
| *SELF* (OURS) | **95.1** | **79.93** | **74.76** | **46.43** |

**Entropy minimization** The typical entropy loss for semi-supervised learning is shown in Fig. 6. It encourages the model to provide extreme predictions (such as 0 or 1) for each sample. Over a large number of samples, the model should balance its predictions over all classes.

The entropy loss can easily be applied to all samples to express the uncertainty about the provided labels. Alternatively, the loss can be combined with a strict filtering strategy, as in our work, which removes the labels of potentially wrongly labeled samples.

For a large noise ratio, predictions of wrongly labeled samples fluctuate strongly over previous training iterations. Amplifying these network decisions could lead to even noisier models model. Combined with iterative filtering, the framework will have to rely on a single noisy model snapshot. In the case of an unsuitable snapshot, the filtering step will make many wrong decisions.

## A.4 MORE EXPERIMENTS RESULTS

### A.4.1 COMPLETE REMOVAL OF SAMPLES

Tab. 8 shows the results of deleting samples from the training set. It leads to significant performances gaps compared to our strategy (*SELF*), which considers the removed samples as unlabeled data. In case of a considerable label noise of 80%, the gap is close to 9%.

Continuously using the filtered samples lead to significantly better results. The unsupervised-loss provides meaningful learning signals, which should be used for better model training.

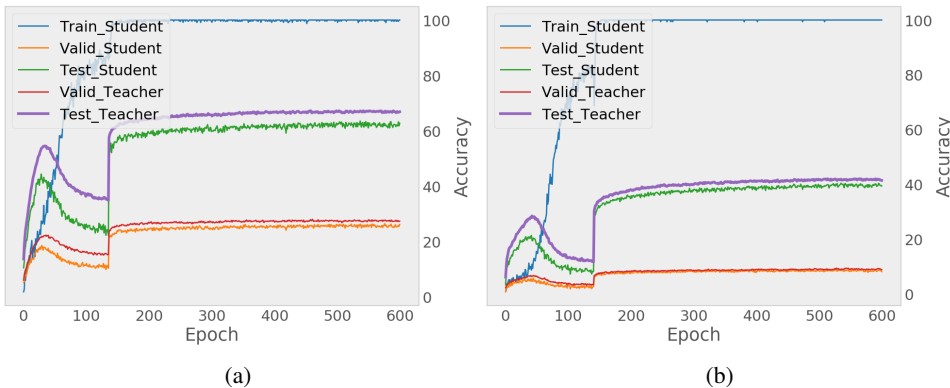

(a)                       (b)

Figure 7: Sample training curves of our approach *SELF* on CIFAR-100 with (a) 60% and (b) 80% noise, using noisy validation data. Note that with our approach, the training loss remains close to 0. Further, note that the mean-teacher continously outperforms the noisy student models. This shows the positive effect of temporal emsembling to counter label noise.

### A.4.2    SAMPLE TRAINING PROCESS

Fig. 7 shows the sample training processes of *SELF* under 60% and 80% noise on CIFAR-100. The mean-teacher always outperform the student models. Further, note that regular training leads to rapid over-fitting to label noise.

Contrary, with our effective filtering strategy, both models slowly increase their performance while the training accuracy approaches 100%. Hence, by using progressive filtering, our model could erase the inconsistency in the provided labels set.

