# OpenReview forum: "SELF: Learning to Filter Noisy Labels with Self-Ensembling"
_ICLR.cc/2020/Conference — Accept (Poster)_

### Official Review · AnonReviewer1 · 2019-10-16
**Official Blind Review #1**

**Rating:** 6

**Review:**

This paper proposed "self-ensemble label filtering" for learning with noisy labels where the label noise is instance-independent (in fact, the noise model is the class-conditional noise). Among the existing directions in this area, it falls into the sample selection direction, but it also takes semi-supervised learning based on the likely noisy data into account.

Novelty: borderline. As other sample selection methods, the proposed one would like to identify the training data with correct labels. What's new is that the authors "form running averages of predictions over the entire training dataset using the network output at different training epochs" and show that "these ensemble estimates yield more accurate identification of inconsistent predictions throughout training than the single estimates of the network at the most recent training epoch". This is the major contribution of the paper. Furthermore, the data likely to have incorrect labels are not thrown away but used in a semi-supervised manner. This is a minor contribution, because semi-supervised learning is orthogonal to label-noise learning and everybody in this area knows the combination of them can work better in practice. Note that this is an academic/scientific paper, not an industrial product, so you don't need to combine all things that might work.

Significance: high. The proposed method significantly outperformed all baseline methods. However, it's not completely fair to compare a label-noise + semi-supervised method with other label-noise only methods... As a matter of fact, you don't need to apply perturbation consistency (or other semi-supervised) regularization after identifying the training data with incorrect labels. Semi-supervised regularization such as virtual adversarial training can even improve supervised learning.

Issues: It's known under class-conditional noise model, the backward loss correction is the unique way to estimate the classification risk (or equivalently, the classification accuracy) given noisy validation data. So how can the validation (i.e., hyperparameter tuning) be performed for the proposed and baseline methods in Table 1 given noisy validation data?

**Experience Assessment:**

I have published in this field for several years.

**Review Assessment: Checking Correctness Of Derivations And Theory:**

I assessed the sensibility of the derivations and theory.

**Review Assessment: Checking Correctness Of Experiments:**

I assessed the sensibility of the experiments.

**Review Assessment: Thoroughness In Paper Reading:**

I read the paper thoroughly.

---

> ### Author Response · Authors · 2019-11-15
> **Thank you for the precious feedback!**
>
> --- Summary-----
> Thank you for providing invaluable feedback. The introduction is updated to highlight the contributions more explicit. The updated Tab. 8 (Appendix) includes semi-supervised learning experiments when combined with recent works. Your questions about other semi-supervised techniques and the validation procedure have led to precious insights: see below (2 and 3).
>
> --- Detailed -----
>
> 1. Contributions:
>
> - As you have acknowledged - the primary contribution is progressive filtering, i.e., form a consensus of predictions (self-ensemble predictions) of the network over different training iterations. The analysis shows that such consensus yield better estimates to progressively filter out the noisy labels.
>
> - The key idea of SELF relies on the self-ensemble predictions of a model. Therefore, it is essential to incorporate a model with stable training behavior to obtain better estimates from the consensus. Concretely, we employ a semi-supervised technique as a backbone to SELF, which provides a stable ground for SELF to retain better consensus throughout learning. Note that this is different from just a mere combination of semi-supervised techniques with a sample selection method.
>
> - The updated Tab. 8 (Appendix) includes additional comparisons in the submission to demonstrate the significance of the SELF-framework by comparing the results against previous works + semi-supervised techniques. The semi-supervised learning provides a robust model consensus to filter out the noisy labels. SELF benefits from this semi-supervised learning more than other techniques (see below).
>
> 2.1. Add semi-supervised learning to baselines:
>
> Thanks for suggesting this comparison! These results have provided additional insights into the SELF-framework. In detail, the updated Tab. 8 (Appendix) shows the in-deep analysis of semi-supervised learning strategies combined with recent works. Our progressive filtering approach is shown to be effective and performs well regardless of the choice of the semi-supervised technique backbone. Overall, the proposed framework SELF outperforms all these combinations.
>
> 2.2. Other semi-supervised regularization can improve supervised learning:
>
> Agreed! However, the focus of this work is to propose self-ensembling as a principled approach to counter label noise. Note that the semi-supervised technique (like Mean-Teacher) is a backbone for our principle. The combination with better semi-supervised techniques is indeed an excellent future direction.
>
> 3. Backward-loss correction is a unique way to estimate classification risk:
>
> - We observe that the risk estimation in SELF is far from optimal. Results from Tab. 1 shows that SELF relying on noisy validation set performs much worse than SELF*  (up to 10% drop in performance at 80% label noise on CIFAR-10). Here, SELF* uses a clean validation set.
>
> - This current work follows the common practice of using such noisy validation data under the literature of learning from noisy labels. A more in-depth analysis of this aspect is worth further investigation.

---

### Official Review · AnonReviewer2 · 2019-10-24
**Official Blind Review #2**

**Rating:** 8

**Review:**

--- Overall ---

This paper proposes an algorithm for learning from data with noisy labels which alternates between updating the model and removing samples that look like they have noisy labels, thereby allowing the training procedure to focus on clean samples. Overall, I found the paper very well-written, the proposed approach reasonable, and the experiments convincing. I have some questions about what assumptions are required for such a procedure to work, but in general, I think this is a strong paper.

--- Major comments ---

1. I found it somewhat unclear how large the methodological contribution was. In particular, has the approach of filtering out samples based on disagreement with predictions from the model been tried before (i.e. the primary contribution is self-ensembling)? When the proposed method includes multiple pieces (Mean teacher + iteratively creating a filtered dataset + self-ensembling), I recommend being *very* explicit about which parts are new contributions. With that said, I greatly appreciated the ablation experiments which really highlight the importance of each piece.

2. I don't feel like I have a good sense for what assumptions need to be satisfied for the proposed method to work. For example, in section 2.2 the authors say "If the noise is sufficiently random, the set of correct labels will be representative to achieve high model performance". What is meant by "sufficiently random" here? Is there a formal version of this assumption? Do any independence or positivity assumptions need to be satisfied? Most importantly, what happens when the label noise does not look like what you expect? I would love to see some experiments examining the failure modes of the algorithm. For example, what happens when label errors are concentrated in a particular region of the feature space (or just generally depend on the features)? In this case, even if the filtering procedure work perfectly, the filtered dataset will have a different feature distribution than the data distribution leading to potential covariate shift problems. If I understood the experiments correctly, the method was only tested on label-depended noise models.

3. Along the same lines: what are the necessary conditions to guarantee that this procedure converges? While the authors suggest that self-ensembling prevents samples from oscillating in and out of training set, is this a guarantee or an empirical observation? More broadly, it is not totally clear what the filtering does to the objective function or whether this procedure is even formally optimizing a well specified objective function (potentially some temperature limit of a soft-weighted objective?).

--- Minor comments ---

1. Figures 1 and 4 are not readable in black and grey-scale.

2. I would front-load the justification for using self-ensembling. In particular, I think the two sentences starting with "When learning under label noise,..." on page 2 could be moved much earlier.

3. I'll be interested to see what the other reviewers say, but I found Figure 2 hard to follow.

4. The formatting of Section 4.2.4 makes it a bit hard to figure out where the text starts (as opposed to the table captions).

**Experience Assessment:**

I have published one or two papers in this area.

**Review Assessment: Checking Correctness Of Derivations And Theory:**

I carefully checked the derivations and theory.

**Review Assessment: Checking Correctness Of Experiments:**

I assessed the sensibility of the experiments.

**Review Assessment: Thoroughness In Paper Reading:**

I read the paper thoroughly.

---

> ### Author Response · Authors · 2019-11-15
> **Thank you for providing fascinating perspective on the proposed framework SELF!**
>
> --- Summary ---
> Thank you for providing interesting perspectives and initiates precious discussions on the proposed method SELF. The updated supplementary materials include the question of required assumptions, failure cases, the model convergence. Additionally, the contributions are made more explicit in the Introduction.
>
> --- Major comments ---
> 1.1. More explicit contributions:
> Thank you for this hint! We completely agree and have modified the description of contributions.
>
> 1.2. Is the sample filtering technique novel:
> Correct, we are the first to employ the disagreement with the model predictions to filter samples progressively.
>
> 1.3. Is the self-ensembling the primary contribution:
> Yes, besides the progressive refinement of the set of correct samples.
>
> 2.1. Required assumptions: Exciting question!
>
> - Natural robustness assumption of deep networks (Rolnick et al. 2017): The networks attempt to learn the easiest way to explain most of the data. SELF uses this assumption to kickstart the process.
>
> - Correct samples dominate over wrongly labeled samples: At 80% noise on CIFAR-10, the correctly labeled cats (20% out of all cat images) still dominates over samples wrongly labeled as cat (8.888% for each class).
>
> - Independence results in less overfitting: SELF performs best if the noises on the validation set and training set are independent and identically distributed. SELF uses the validation data for early stopping. Hence, a high correlation of label noise between train and valid increases the chance of model overfitting.
>
> - Sufficient label noise randomness assumption: see below (2.2)
>
> 2.2. What does sufficiently random label noise mean:
>
> - Sufficient label noise randomness assumption: Each class consists of multiple clusters. If one distinct cluster is entirely covered by label noise, SELF will struggle to expand the set of correct samples to cover this cluster.
>
> - SELF performs progressive expansion of the correct labels sets. At larger noise ratios, not all clusters are covered by the identified correct samples. Therefore on task containing many classes, e.g., CIFAR-100, the model performance decreases faster than on CIFAR-10.
>
> 2.3. Failure cases:
> When the assumptions from 2.1. are strongly violated. Each assumption should have its own "critical" threshold for violation. A future in-depth analysis to challenge the assumptions is fascinating.
>
> 2.4. Labels noise only in a particular region or depend on the features:
>
> - The asymmetric noise on CIFAR-10 closely resembles this scenario. Concretely, the class plane might be randomly flipped to the class ship. Most samples should share similar features within the class plane.
>
> - Adding label noise to more fine-grained samples regions is interesting. However, coherent regions in a learned feature space might depend strongly on the network architectures used to learn on the clean datasets. Hence, the simulation or acquisition of these noisy types is not trivial.
>
> 2.5. Only tested on label-dependent noise:
> Uniform (symmetric) label noise is label-independent. Sample from each class is randomly flipped uniformly to all classes.
>
> 3.1. Convergence of SELF: We have not observed any model divergence in the experiments.
>
> - The natural robustness assumption (2.1) needs to hold to kickstart the process. In SELF, the model learns the easiest way to explain most of the data.
>
> - As the progressive filtering proceeds, the noise ratio is slowly reduced. So the learning task becomes significantly easier. The always-on unsupervised objective compensates for the small sample size induced by filtering on all samples.
>
> 3.2. Guarantee for oscillation of wrong labels:
> We argue that no guarantee exists for the oscillation of all wrong samples. Some samples might resemble the wrong class more than their actual class. Depending on the perspective, the tiger might look more like a cat than other tigers.
>
> 3.3. Is the new objective function well-specified:
> The learning objective in SELF adds to the standard loss two aspects: the model remains close to its self-ensemble and progressive filtering of noisy labels. The self-ensemble learning is well-defined. The progressive filtering is also robust and do not add any instabilities to the training process.
>
> --- Minor comments ---
> - Fixed or will be fixed in the final version.
> - Front-load the justification for using self-ensembling: Great hint! Updated the Introduction accordingly.

---

### Official Review · AnonReviewer3 · 2019-10-27
**Official Blind Review #3**

**Rating:** 3

**Review:**

Summary:
The paper proposed a self-ensemble label filtering (SELF) method to deal with the noisy label learning problem. They progressively filter out the wrong labels during training, i.e.,  filtered samples are removed entirely from the supervised
training loss, and are leveraged via semi-supervised learning in the unsupervised loss. The filtering is based on identification of inconsistent predictions throughout training.

Strengths:
1. The motivation of the paper is very clear.
2. Experiments are conducted on various dataset CIFAR10, CIFAR-100 and ImageNet.

Weakness:
1. The contribution of SELF is not clear. Just a combining of several previously proposed components?
2. For the experimental comparisons, the authors at least should report the acc on clean test set, which is useful for understanding the ideal case performance.
3. The organization of the tables and figures are somehow hard to read.
3. The comparisons are not fair. SELF incorporate semi-supervised techniques while baselines are not.
4. The author missed some important baselines here.
     1) Symmetric cross entropy for robust learning with noisy labels, ICCV2019
     2) Joint Optimization Framework for Learning with Noisy Labels, CVPR2018
     3) Dimensionality-driven learning with noisy labels, ICML2018

**Experience Assessment:**

I have published in this field for several years.

**Review Assessment: Checking Correctness Of Derivations And Theory:**

I carefully checked the derivations and theory.

**Review Assessment: Checking Correctness Of Experiments:**

I carefully checked the experiments.

**Review Assessment: Thoroughness In Paper Reading:**

I read the paper thoroughly.

---

> ### Author Response · Authors · 2019-11-15
> **Thank you for your suggestions! The additional results provide valuable insights into our method SELF.**
>
> Thank you for your great feedback. The contributions are now more explicit. The updated Tab. 8 (Appendix) contains semi-supervised learning experiments in combination with recent works. Thank you for your suggestions! The additional results provide valuable insights into our method SELF.
>
> 1. Clarified contributions:
>
> - The method SELF is a simple but effective mechanism to filter out noisy labels during training and exclude wrong labels from supervision. Prior works on semi-supervised learning (Laine and Aila, 2016; Luo et al., 2018) suffer from the influence of label noise on the supervised loss and thus influence the task performance.
>
> - To the best of our knowledge, we are the first to identify and propose self-ensembling as a principled technique to learn robustly. Other self-ensembling techniques, such as multiple checkpoints or model restarts, are also applicable. Self-ensembling works because the predictions of wrongly labeled samples often oscillate over training iterations.
>
> - As you have correctly acknowledged - the primary contribution is progressive filtering, i.e., form a consensus of predictions (self-ensemble predictions) of the network over different training iterations. The analysis shows that such consensus yield better estimates to progressively filter out the noisy labels.
>
> - The key idea of SELF relies on the self-ensemble predictions of a model. Therefore, it is essential to incorporate a model with stable training behavior to obtain better estimates from the consensus. Concretely, we employ a semi-supervised technique as a backbone to SELF, which provides a stable ground for SELF to retain better consensus throughout learning. The updated in-depth analysis of semi-supervised learning confirms this effect. More follows in 4. Note that this is different from just a mere combination of semi-supervised techniques with a sample selection method.
>
> 2. Add ideal accuracy on the clean test set:
> Great suggestion! The updated Tab. 3 include the baseline accuracy of networks that are trained in a fully supervised setting at 0% label noise. All performances are reported on the clean test set.
>
> 3. Reorganize tables and figures:
> The tables and figures are rearranged to improve readability.
>
> 4. Add semi-supervised learning to baselines:
> Thanks for suggesting this comparison! These results have provided additional insights into the SELF-framework. In detail, the updated Tab. 8 (Appendix) shows the in-deep analysis of semi-supervised learning strategies combined with recent works. Our progressive filtering approach is shown to be effective and performs well regardless of the choice of the semi-supervised technique backbone. Overall, the proposed framework SELF outperforms all these combinations.
>
> 5. More baselines:
> Thanks for making us aware of these highly relevant previous works: SL (Wang et al., 2019), JointOpt (Tanaka et al., 2018), D2L (Ma et al., 2018). The updated Tab. 1 and Tab. 2 include these baselines. The  SELF-framework consistently outperforms these methods.

---

> > ### Comment · AnonReviewer3 · 2019-11-15
> > **About the results in Table 1 & 2...**
> >
> > Are the results in Tables 1 & 2 in the manner copy and paste for baselines?

---

> > > ### Author Response · Authors · 2019-11-15
> > > **Clarification**
> > >
> > > Thank you for asking!
> > >
> > > We implement JointOpt (Tanaka et al., 2018) based on the official implementation.
> > > For SL (Wang et al., 2019) , D2L (Ma et al., 2018), we adopt the performance from the respective publication.

---

### Public Comment · ~ning_mou1 · 2019-11-08
**Convergence issues**

The ensemble predictions given by the teacher model can also be wrong, thus the selected images can be noisy. At extreme noise levels, say 90%, the model is not guaranteed to converge. It may diverge since most of the predictions are inconsistent and the selected images are few and noisy. What do you think about it?

---

> ### Author Response · Authors · 2019-11-15
> **We observe that our model always converges, even at extreme noise ratios.**
>
> - The most critical step is the first training step on the initial label set.
>
> - Due to the natural robustness of deep networks [Rolnick et al. (2017)], the model can still be improved by training and hence do not diverge. The intuition is that deep networks attempt to learn the "easiest" patterns. For instance, at 80% noise on CIFAR-10, the correctly labeled cats (20% out of all cat images) still dominates over samples wrongly labeled as cat ($8.\bar{8}$ $\%$ for each class). In this cat example, the model learns the features to explain the largest numbers of samples. Hence, the model learns important cat features to explain 20% of the samples.
>
> - As the progressive filtering proceeds, the noise ratio is slowly reduced.  The learning on the refined label set is easier for the model. The unsupervised objective compensates for the reduction of the data set size.  Consequently, the progressive filtering help model convergence.
>
> - Empirically, we observe that our model always converges, even at extreme noise ratios.
>
> - Note that 90% noise on CIFAR-10 corresponds to completely random labels. There exists no model which is better than random. Consequently, the critical limit on CIFAR-100 is at a 99% noise ratio (uniform noise).

---

### Public Comment · ~Pengfei_Chen1 · 2020-03-16
**Request for Code Release**

Thank you for your excellent work. I would like to reproduce the results. Would you consider releasing the code of your method SELF for easy reproduce?

---

> ### Author Response · Authors · 2020-03-17
> **Pending approval**
>
> Hello, Pengfei,
>
> Thank you very much for your interest in our work!
> We have started the approval procedure for the release of the code.
> However, we are not sure if it will be successful soon.
> We will keep you informed as soon as things change.
>
> Best,

---

### Public Comment · ~Gladis_Ne_Limes1 · 2023-06-13
**re**

A dedicated development team refers to a group of professionals who are exclusively assigned to work on a specific project or set of projects within an organization. This team is typically composed of developers https://mlsdev.com/blog/hire-a-development-team, designers, testers, project managers, and other specialists required for the successful execution of software development initiatives.

---

### Decision · Program_Chairs · 2019-12-19

**Decision:**

Accept (Poster)

**Comment:**

The authors addressed the issues raised by the reviewers; I suggest to accept this paper.